# National medicines regulatory authorities financial sustainability in the East African Community

**Margareth Ndomondo-Sigonda**[1,2]*, **Jacqueline Miot**[3], **Shan Naidoo**[4], **Brian Ng'andu**[2], **Nancy Ngum**[2], **Nelson E. Masota**[5,6], **Eliangiringa Kaale**[6]

1 Pharmacology Division, Department of Pharmacy and Pharmacology, Faculty of Health Sciences, University of Witwatersrand, Johannesburg, South Africa, 2 African Union Development Agency -New Partnership for Africa's Development (AUDA-NEPAD), Midrand, South Africa, 3 Department of Internal Medicine, Health Economics and Epidemiology Research Office, School of Clinical Medicine, Faculty of Health Sciences, University of Witwatersrand, Johannesburg, South Africa, 4 Public Health Medicine Department, School of Public Health, University of Witwatersrand, Johannesburg, South Africa, 5 Institute for Pharmacy and Food Chemistry, Julius-Maximilians- University of Wuerzburg, Würzburg, Germany, 6 School of Pharmacy, Muhimbili University of Health and Allied Sciences, Dar es Salaam, Tanzania

* margarets@nepad.org

**Data Availability Statement:** All relevant data are within the manuscript and its Supporting Information files.

## Abstract

### Introduction

Adequate and sustainable funding of national medicine regulatory agencies (NMRAs) is key for assurance of quality, safety and efficacy of medical products circulating in a market. The study aimed to determine factors affecting NMRAs funding in five East African Community (EAC) countries namely: Burundi, Kenya, Rwanda, Tanzania (Mainland and Zanzibar) and Uganda.

### Methodology

An exploratory, mixed method design using both qualitative and quantitative data, was employed. Data from six NMRAs was collected through a combination of semi-structured interviews, questionnaires, and checklists for the period 2011/12-2014/15 while 2010/11 data served as baseline. Interviews were conducted with heads of NMRAs and monitoring and evaluation experts of the respective agencies. NMRA's financing was assessed using six indicators namely, funding policy, financial autonomy, the total annual budget, actual funding per annum, funds received from various sources, and the NMRA expenditure.

### Results

The average total annual budget for all the EAC countries during the study period 2011–2015 ranged from USD 824,328.67 to USD 10,724,536.50. The low budget in Zanzibar may be attributed to population and pharmaceutical market size. Uganda's attainment of 98.75% (USD 10,656,704) revenue from industry fees is a result of deliberate government policy change from 100% reliance on donor funding over a period of 10 years (1995–2015). On average, the proportion of revenue against budget per annum is 54.8% (USD 458,970.11),

**Funding:** The study on which this manuscript is based was funded by AUDA-NEPAD. The AUDA-NEPAD had no input into the study design, data collection, analysis or the preparation of the manuscript.

**Competing interests:** The authors have declared that no competing interests exist.

98.7% (USD 10,302,295.25) and 100% (USD 7,375,802.08) for Zanzibar Food & Drugs Agency (ZFDA), Uganda National Drug Authority (NDA) and Tanzania Medicines and Medical Devices Authority (TMDA) respectively. Governments, industry fees and donors are the major sources of funding across all NMRAs in the EAC region, with TMDA and Uganda NDA relying more on industry fees by 73.20% (USD 4,664,777.59) and 98.25% (USD 8,077,238.20) respectively. While Burundi relies solely on government funding, ZFDA, on the other hand, received on average 50.40% (USD 252,557.22) from government and 40.60% (USD 165,303.34) from industry fees and the remaining 9% from donors and other sources. An overall contribution of funds received from donors by each NMRA was the least among other sources of financing. Observation of expenditure patterns indicated operational costs to be the major expense in the majority of the NMRAs, followed by salaries and infrastructure development. The Kenya NMRA has the highest degree of average expenditure across all three categories, with the least average expenditures being marked by Burundi NMRA. The operational costs on average increased considerably in all the NMRAs during the study period.

## Conclusion

Evidence from the EAC suggests that government and industry fees are the main sources of funding while donor contributions vary from country to country. Government policy, legal framework, and fees structure are the key enablers of NMRAs funding sustainability.

## Introduction

National Medicines Regulatory Agencies (NMRAs) are responsible for carrying out a number of regulatory functions including: registration and marketing authorization, vigilance, market surveillance and control, licensing establishments, laboratory testing, clinical trials oversight and NMRA lot release, just to mention a few [1]. All these activities require adequate financial resources to assure the quality, safety and efficacy of medical products on the market and promotion of patient safety as a critical part of the health care delivery system [2]. Studies show that this important task of regulating medical products is often under-funded and under-recognized in many countries in Africa [2].

Sustainable funding is one of the key factors to ensure effective regulation of medical products, others include a comprehensive legal framework, appropriate and adequate governance mechanisms, and sound technical expertise [3]. The legal basis gives the NMRA power to perform a function. However, there are a number of prerequisites to its performance. These include, the level of autonomy in executing its mandate, the appropriate structure that allows for proper coordination of various regulatory activities, availability of financial resources and adequate number and type of appropriately skilled human resources with requisite competency to carry out their duties [4].

Sustainability of financial resources for NMRAs means having a specific budget assigned to medicines regulation and assurance that the allocated funds are safeguarded against the competing needs of other government agencies [4]. NMRAs funding sources can be derived from public funding, fees for services provided and donations to supplement the often limited funding available from government [5,6]. The stability of an NMRA depends on its financing mechanism [6].

Globally, medicines regulators struggle with the need for financial and technical resources to fully meet their mandate. Generally, NMRAs from well-resourced countries such as the United States of America have reliable funding. For instance, the US Food and Drug Administration (US-FDA) budget for 2019 financial year is estimated at USD 5.7 billion, 55% of which (USD 3.1 billion) is provided by the federal government and the remaining 45% (USD 2.6 billion) is paid for by industry user fees [7]. In the United Kingdom, the Medicines and Healthcare products Regulatory Agency (MHRA) employs mixed funding arrangements where the Department of Health and Social Care) (DHSC) funds regulation of medical devices, whilst the costs of medicines regulation is met through fees from the pharmaceutical industry [8]. Studies conducted in Sub-Saharan Africa, on the other hand, have shown that in nine out of the twenty six States studied (34.6%) the NMRAs depend on government funding with all fees paid directly to treasury and not redistributed. In addition funds allocated by the States to their respective NMRAs are not released on time [5].

Due to limited publications on sustainable financing models for NMRAs, especially in low-middle income countries, this study is intended to contribute to the existing knowledge on the various funding sources for NMRAs, factors affecting sustainability and to propose sustainable funding options using the East Africa Community (EAC) as a case study.

## Methodology

An exploratory, mixed method design using both qualitative and quantitative data, was employed in this study. Data was collected from all the six NMRAs in the EAC partner States namely, the Republic of Burundi, Republic of Kenya, Republic of Rwanda, the United Republic of Tanzania (with two NMRAs), and the Republic of Uganda. Data from six NMRAs was collected through a combination of semi-structured interviews, questionnaires and checklists as indicated in S1 File, for the period 2011/12-2014/15 while 2010/11 data served as baseline.

Questionnaire and checklist were developed based on the information obtained from the preliminary situational analysis study. Moreover, additional questions were adopted from the WHO Global benchmarking tool for evaluation of national regulatory systems [1]. Validation of the questionnaire and checklists was done using a pilot study.

The first phase of data collection involved self-administration of the questionnaire and checklists (S1 File) by the selected informants. These included the NMRA's head, one monitoring and evaluation personnel from each NMRA and a project officer from the EAC MRH project, making a total of 13 respondents. All informants were purposefully selected based on their roles and participation in medicines policy and regulation of medical products in the respective NMRA.

Semi-structured interviews, also involving the above-mentioned respondents, were conducted following the successful completion of the questionnaire and checklists data collection phase. In addition, one NMRA staff responsible for medicines registration, GMP inspections, legal affairs, human resource and finance were also interviewed in each NMRA. An invitation letter and interview topic guide were sent to the interviewees in advance. Interviews were conducted on face to face basis, each session lasting for 1 to 2 hours. Responses were recorded by means of selective written notes, which were thereafter subjected to qualitative analysis. Follow up visits were conducted aiming at collecting the missing data and validating the previously collected data.

Due to lack of data from some agencies, only the available data set was used in statistical analysis and interpretation. The prevailing average annual USD currency exchange rate for each country was used for analysis. The average annual exchange rates of Tanzanian, Kenyan, Ugandan, Rwandese and Burundian currencies against the USD ranged as TZS (1585.3–

2188.2), KES (84.2–101.5), UGX (2349.4–3369.4), RWF (579.9–783.1) and BIF (1159.7–1657.6) respectively over the studied period [9].

NMRAs financing was assessed using six (6) indicators namely: NMRA financing policy, level of NMRA autonomy, the total annual budget for carrying out regulatory functions, actual funding per annum, funds received from various sources, and the NMRA expenditure as shown under indicators guidance notes S2 File.

Quantitative data were analysed for means and standard deviation using Microsoft Excel©, whereas document analysis was used to extract organize and interpret data from policies and laws. This study was part of a larger overall evaluation of the NMRAs implementing medicines regulatory harmonization program in the EAC.

### Ethical clearance

Ethics approval was granted by the WITS Human Research Ethics Committee on 31[st] July 2015 through clearance certificate No. M150751. In addition, national ethical clearance was granted by respective national research ethics committees and ministries of health. Informed consent forms were signed by individual respondents from NMRAs in the EAC partner states.

## Results

### Funding policies and laws of NMRAs

All six NMRAs provided data on their national medicines policies (NMPs) and medicines laws. While all the NMPs emphasise the need for effective regulation of medical products, policy commitment on funding NMRAs varies across countries. For semi- or fully autonomous NMRAs such as the Tanzania Medicines and Medical Devices Authority (TMDA), the National Drug Authority in Uganda (NDA), the Pharmacy and Poisons Board in Kenya (PPB), the Rwanda Food and Drugs Authority (Rwanda FDA) and the Zanzibar Food and Drug Agency (ZFDA), the laws provide for collection of fees from industry and its utilization to perform regulatory services [10–14]. However, in Burundi, the NMRAs was observed to be fully reliant on government funding during the studied period [15].

### Level of NMRA financial autonomy

There has been a progressive transformation of the level of financial autonomy of NMRAs in the EAC region over the study period (Table 1). While Burundi NMRA is not autonomous, the Bill to transform the existing unit to an autonomous agency is at an advanced stage of consideration by her parliament [16]. In Rwanda on the other hand, the Food and drugs act of 2013 [11] has been enacted to establish the Rwanda Food and Drugs Authority (Rwanda FDA) as a semi-autonomous Agency. In Zanzibar, the former Zanzibar Food and Drugs Board (ZFDB) has been transformed into Zanzibar Food and Drugs Agency (ZFDA) by the Zanzibar Food, Drugs and Cosmetics Act number 2 of 2006 [13] with financial autonomy under the oversight of an Advisory Board. The existing laws in Kenya, Uganda and Tanzania-Mainland, provide for financial autonomy and empower the respective NMRAs to collect and utilise fees for services rendered [10,12,14].

### NMRAs annual budgets

The average total annual budget for all the countries for the period 2011–2015 ranged from USD 824,328.67 to USD 10,724,536.5 as shown in Fig 1, with Zanzibar and Uganda NMRAs having the lowest and highest average total annual budgets respectively. The NMRAs in Tanzania and Kenya exhibited almost similar average annual budgets, while the data for this

**Table 1. Level of NMRA autonomy in the EAC partner states between the years 2011 and 2015.**

| Indicator | National Medicines Regulatory Agency | | | | | |
|---|---|---|---|---|---|---|
| | **Burundi** | **Kenya** | **Rwanda** | **Tanzania** | **Uganda** | **Zanzibar** |
| NMRA's level of autonomy | Department within the Ministry of Health | Semi-Autonomous | Semi-Autonomous | Semi-Autonomous | Autonomous | Semi-Autonomous |
| Medicine Law | Republic of Burundi. Decret No. 100/150 du 30 September 1980 portant Organization de l'exercice de la Pharmacie au Burundi. 1980. | Republic of Kenya. The Pharmacy and Poisons Act, Chapter 244. 1957, as amended in 2009. | Republic of Rwanda. Law No. 47/2012 of 14/01/2013 relating to the Regulation and Inspection of Food and Pharmaceutical Products. 2013 | United Republic of Tanzania (Mainland). Tanzania Food, Drugs and Cosmetics Act, Cap 219. 2003, as amended in 2004, 2014 & 2019 | Republic of Uganda. The National Drug Policy and Authority Act. 1993. | United Republic of Tanzania (Zanzibar). The Zanzibar Food, Drugs and Cosmetics Act. No. 2 of 2006 as amended in 2016 |

indicator was not available for Burundi's and Rwanda's NMRAs. When compared to their own baselines, the total annual budget of each NMRA approximately doubled over a duration of five years.

## NMRAs annual funding

Comparison of the total annual budget against actual funding per annum shows an average of 54.8% (USD 458,970.11), 98.7% (USD 10,302,295.25) and 100% (USD 7,375,802.08) for Zanzibar Food & Drugs Agency (ZFDA), Uganda National Drug Authority (NDA) and Tanzania Medicines and Medical Devices Authority (TMDA) respectively, for years 2011–2015. However, data from Kenya and Rwanda under this aspect was not available (Fig 2). The 54% funding for ZFDA may entail either a problem with setting an unrealistic budget or simply that funding was inadequate to support NMRA activities.

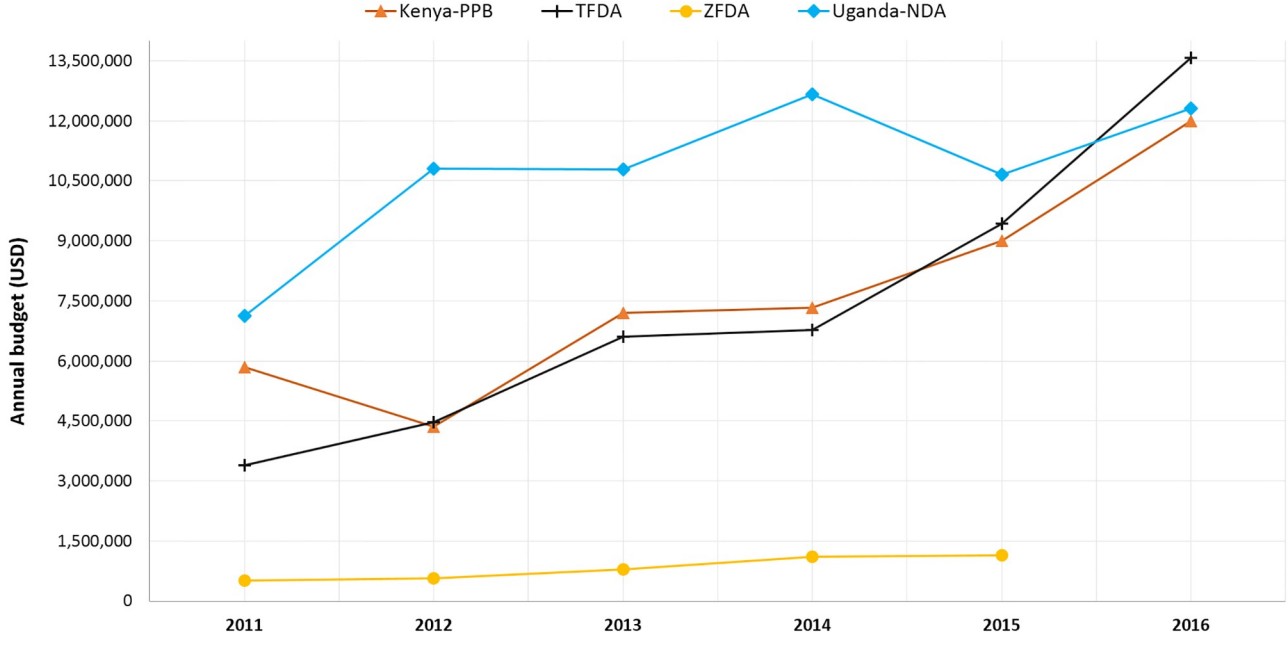

**Fig 1. Trends in East African Community national medicines regulatory authorities' annual budgets.**

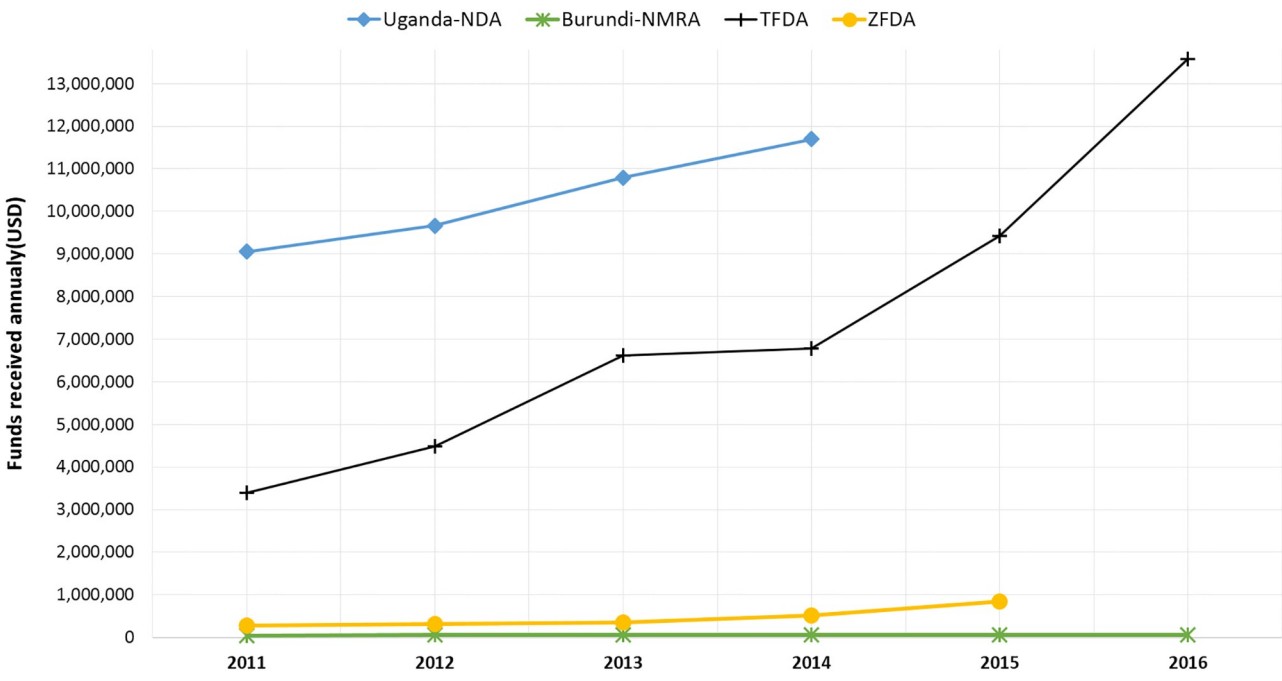

**Fig 2. Trends in annual funding among national medicines regulatory authorities in the East African Community.**

## Sources of revenue for NMRAs

Assessment of the various sources of funds received by the NMRAs indicated governments, industry fees, and donors to be the major funding sources across all NMRAs as shown in Fig 3. It should be noted that the PPB in Kenya only provided data on annual budgets without indicating the sources of its revenue. For the NMRAs in Tanzania-mainland (TMDA) and Uganda

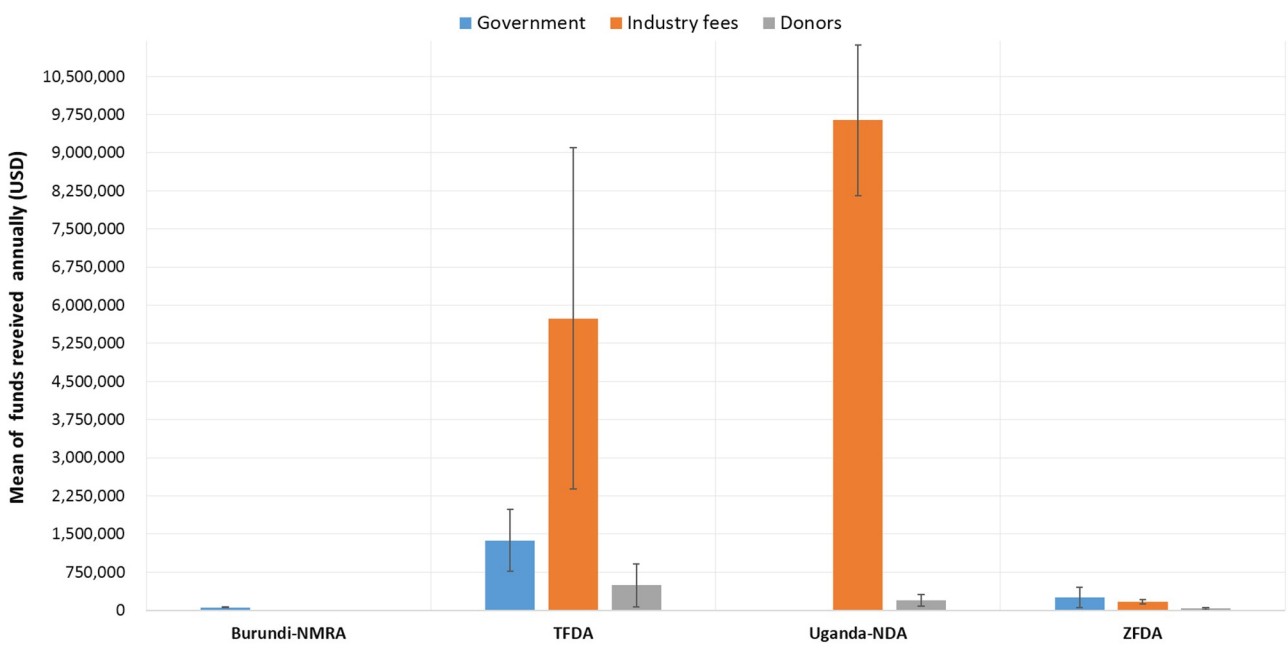

**Fig 3. Annual mean amounts and sources of funds received by National medicines regulatory authorities.**

(NDA), industry fees were observed to be the major source of funds. On average, industry fees contributed up to 73.20% and 98.25% of the total annual revenue for TMDA NDA respectively. For Zanzibar (ZFDA) and Burundi NMRAs, the governments were observed to be the main sources of revenue with Burundi relying solely on the government funding while for ZFDA, on average government contributes 50.40%, industry fees 40.60% and the remaining 9% of the total revenue is from donors and other sources.

Funds received from donors indicated a high degree of variation, with some NMRAs receiving funding only once over a five years duration, while others received fluctuating amounts with no remarkable trends. Funding partners most cited include the Bill and Melinda Gates Foundation (BMGF), World Health Organization (WHO), United Nations Industrial Development organization (UNIDO), Clinton Health Access Initiative (CHAI), United Nations International Children's Fund (UNICEF), Management Sciences for Health (MSH), World Bank, Trademark East Africa Limited (TMEA), German Corporation for International Cooperation (GIZ), Global Fund, the United States Pharmacopoeia (USP) and the United States Agency for International Development (USAID) just to mention a few. On average, the overall contribution of funds received from donors by each NMRA was the least among other sources of financing.

## NMRAs' expenditure

Observation of expenditure patterns indicated operational costs to be the major expense in the majority of the NMRAs, followed by salaries and infrastructure development as shown in Fig 4. This is also explained by the observed increase in the number of staff among the NMRAs from which the annual number of staff could be obtained (Fig 5). The PPB in Kenya indicated the highest degree of average expenditure across all three categories, with the least average expenditures being marked by the Burundi NMRA. Data from Rwanda and Zanzibar NMRAs were not available for this indicator. The operational costs on average increased considerably across all the NMRAs during the studied period.

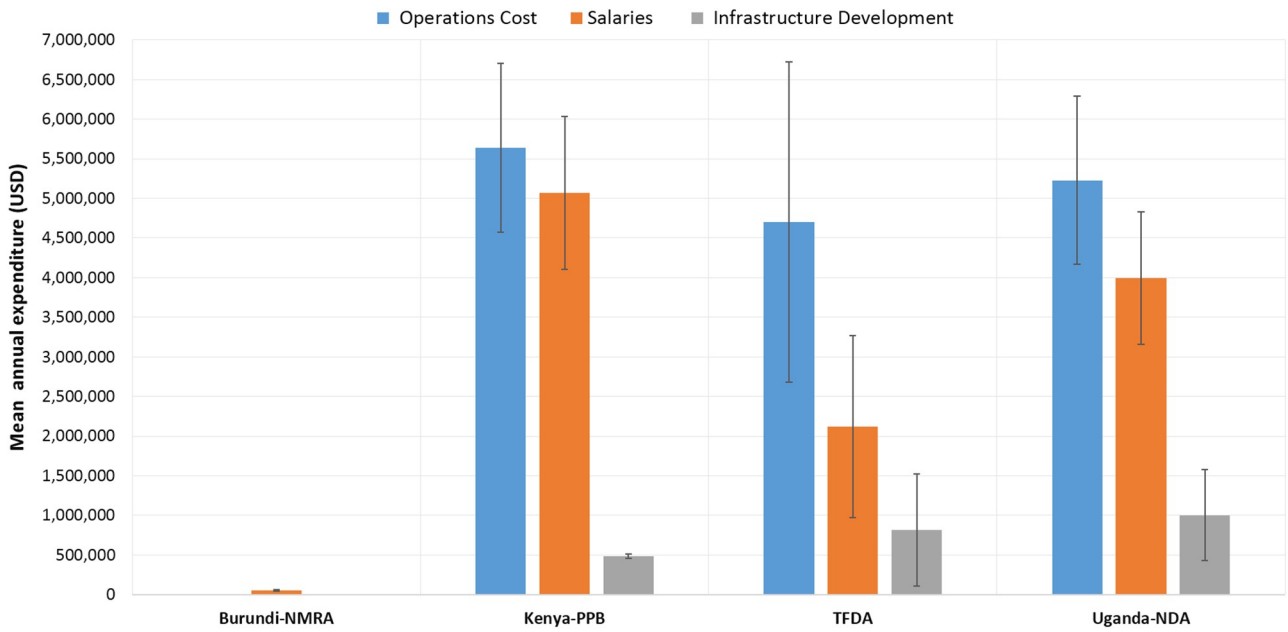

**Fig 4. Mean annual amounts and main areas of expenditure by the national medicines regulatory authorities.**

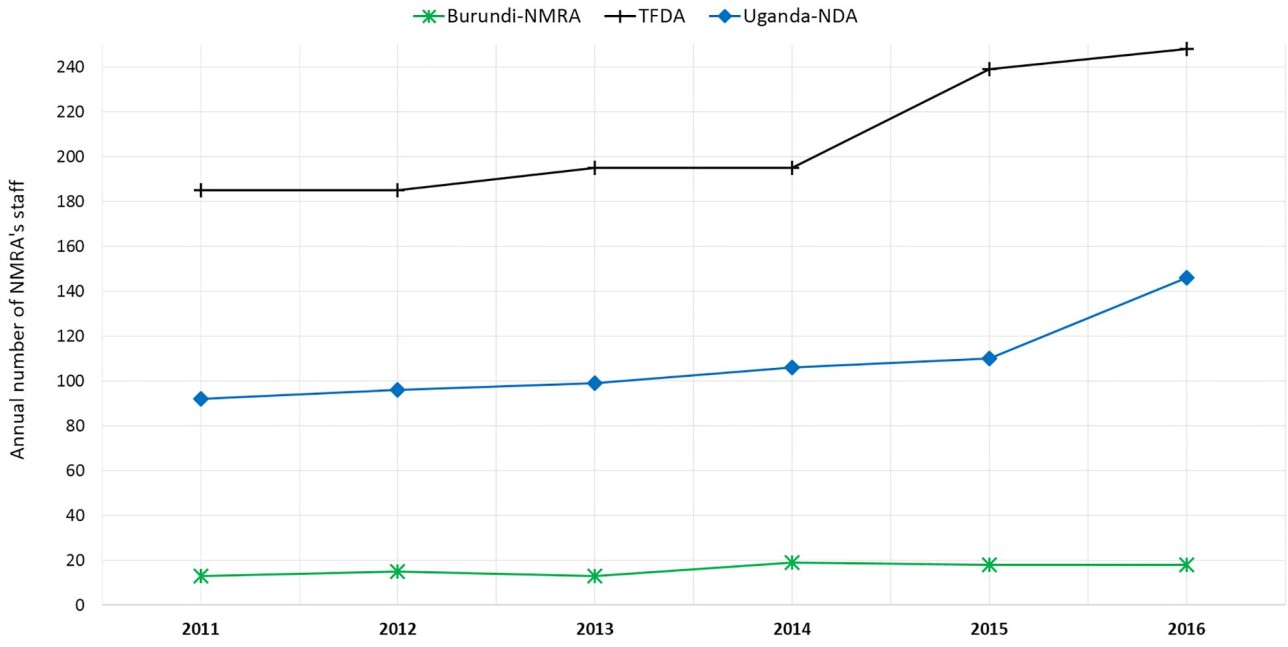

**Fig 5. Annual trend in the number of staff per national medicines regulatory authority.**

## Discussion

Clear government policy and legal framework to empower the NMRAs to collect and utilise fees for services offered is an essential element for sustainable funding [4]. Results from this study show that governments across the EAC region use different NMRA financing models to regulate pharmaceutical markets. While some rely entirely on government funding as is the case for Burundi NMRA, others use a combination of sources of revenue from government (50.40%), industry fees (40.60%) and donors and other sources (9%) as exemplified by ZFDA out of an average annual budget of USD 824,328.67. In the case of TMDA with the annual budget increasing from USD 3,384,123.00 to USD 9,422,888 and the contribution of industry fees to the total budget increasing over the years from 60% (USD 2,018,608.88) in 2011 to 86% (USD 8,123,093) in 2015, while the contribution from government has been steady at an average of 19.60% (USD 1,168,299.09) over the studied period. Yet, the government of Uganda largely depends on industry fees (98.25%, USD 8,077,238.20) to finance the NDA activities. The increase in revenue could be associated with the observed improvement in registration processes following the harmonization of registration systems and the introduction of quality management systems among the EAC Partner States [17].

Findings from this study are in agreement with the WHO multi-country study conducted in 2002 [4]. The WHO study reported a decrease from 100% to 60% donor funding between 1995 and 1997 in Uganda as the result changes in the NMRA funding policy, with the government and industry fees making up only 20% each. Currently, the Uganda NDA is 98.25% (USD 8,077,238.20) funded through fees for service with minimal contribution from donors. While there is no enough data for comparison with other EAC countries, the findings support the need for sound government policies and legal frameworks to empower the NMRA to collect and utilise fees as means to ensure financial sustainability [4].

In terms of an enabling environment for fees collection by the NMRAs, while the NMRAs in Kenya, Tanzania, Uganda, Rwanda, and Zanzibar have legal mandate to collect and utilise

fees, Burundi does not. Considering that medical products regulation is a public function, safeguarding funding for regulation of medical products against the competing needs of other government agencies is key for sustainability of NMRAs.

An alternative funding model is a combination of government budgetary allocation and industry fees, whereas the collected fees are transferred to the government central treasury as is the case for Malaysia and Venezuela [4]. This is in line with the study among Sub-Saharan African countries, which revealed that 35% of the NMRAs depend on government funding, with all fees paid directly to Treasury [5]. It has also been reported that the fees and charges are set arbitrarily and not necessarily linked to the cost of service provided while resource intensive services are offered free [4]. Therefore, for this model to be sustainable, the existing fees and other charges should reflect the real cost of services provided. Under such conditions, the NMRA can rely entirely on the charged fees to finance regulatory activities. In a situation where the collected fees are transferred to the government central treasury, the government must ensure that the funding allocation meets the budget requirement. While it is important that NMRAs are adequately financed to deliver on their mandate, attention should also be given on NMRAs expenditure to ensure accountability and balanced utilization of funds for public interest [4].

Another model is where the fees and charges reflect the real cost of services provided such as evaluation of dossiers and inspection of establishments, and the NMRA is financed entirely on fees for service as is the case with the Therapeutic Goods Administration (TGA) in Australia and Medicines Control Authority of Zimbabwe (MCAZ) which have full powers to use the revenue they collect [4]. The TGA in Australia is an example of an agency that moved gradually from government-financed to a self-financed agency through a government policy which was phased in over a five years period from 1994–1999. Industry fees, therefore, provide a reliable and sustainable source of revenue for NMRAs.

This is especially so as Africa witnesses convergence of changing economic profiles, rapid urbanisation, increased healthcare spending and investment, and increasing incidence of chronic diseases which is attributed to the USD 45 to 60 billion projection of the African pharmaceutical market by 2020 [6]. The increasing demand for medicines in Africa including the EAC region attributed by population and economic growth as well as raising consumer awareness warrants governments' investment on regulation of medical products [18].

Furthermore, governments investment in regional harmonization efforts and strengthening regulatory capacity and systems across NMRAs in the EAC partner states has a significant impact in improving availability of medicines through timely marketing authorization of essential medicines, reduced duplication of individual NMRAs efforts, streamlined use of limited resources with resultant savings in public health budgets [19,20]. This is exemplified by the EAC medicines regulatory harmonization program which has increased efficiency in registration processes among the NMRAs in the region with a subsequent reduction in registration timelines from the previous 1–2 years period to a median of 7 months [21].

In recognition of the need to ensure that all African citizens have access to safe, quality and efficacious essential medicines, the African Union approved the African Medicines Regulatory Harmonization (AMRH) Initiative as a key pillar of the Pharmaceutical Manufacturing Plan for Africa (PMPA) [22]. The AMRH Initiative which covers more than 85% of Sub-Saharan Africa serves as a foundation for establishment of the African Medicines Agency. AMA will build on the AMRH success through coordination of on-going regulatory systems and strengthening and harmonization efforts of the AU, RECs, Regional Health Organization (RHOs) and member States [23]. AMA also provides a good platform to support the ongoing African Continental Free Trade Area (AfCFTA)—anchored pharmaceutical project for the Small Island States (SIDS) and land locked countries including Seychelles, Madagascar,

Comoros, Mauritius, Djibouti, Eritrea, Rwanda, Ethiopia, Kenya, Sudan and IGAD. This pilot health and economic initiative established through public-private partnerships and innovative financing is intended to contribute to improved accessibility and affordability to safe medicines and to accelerated progress towards SDGs and Agenda 2063. All these initiatives provide an opportunity to invest in medicines regulation on the African continent.

## Limitations of the study

In this study, data on population size, the number of pharmaceutical establishments and the pharmaceutical market size was intended to determine the level of investment (in terms of financial and human resources) required to control the market. However, only Burundi provided data on pharmaceutical market size making analysis and comparison between countries difficult.

It is also worthwhile noting another limitation in this study as a lack of information on the actual NMRAs fees structure which provides different streams of fees charged and how the funds collected are distributed in performing the different regulatory functions. A further study on fees charged by different NMRAs will provide useful information on fees structure and its basis as well as proportion of contribution from various streams of revenues.

## Conclusion

Inadequate funding for NMRAs is one of the major challenges hampering effective regulation of medical products world-wide and within the East African Community. Measures are needed to guide countries to institute appropriate policies and legal frameworks which will ensure that NMRAs are empowered to collect and utilise fees for services they offer. Where the NMRA is not fully mandated and the market size does not provide enough financial resources for the NMRAs to perform their functions effectively, governments must ensure a dedicated budget is allocated to facilitate NMRA performance. Governments should also invest in NMRAs participation in regional harmonization efforts which have proven to facilitate effective and efficient utilization of already limited resources while at the same time reducing the time taken for applicants to put the product on the market.

There is a need to conduct further research to assess the fee structure employed by various NMRAs to determine whether it reflects the actual cost of service or not. The study can also explore the proportion of contribution from various streams of revenue and how they are allocated to support various regulatory functions and NMRAs participation in regional harmonization efforts. In addition, a study correlating population, pharmaceutical market size, with the level of investment needed to ensure effective market control needs to be conducted. This will assist in guiding governments on the level of investment needed for the NMRA in terms of infrastructure, financial and human resources, to ensure effective regulation of pharmaceutical market in a country.

## Supporting information

**S1 File. Questionnaire and checklist for heads of NMRAs and M&E experts.**
(PDF)

**S2 File. Indicators guidance notes.**
(PDF)

**S1 Table. Data.**
(PDF)

## Acknowledgments

Authors would like to express their sincere gratitude to the officials in the NMRAs of Tanzania, Kenya and Uganda are sincerely appreciated for their good support during the data collection and conduct of this study.

## Author Contributions

**Conceptualization:** Margareth Ndomondo-Sigonda, Jacqueline Miot, Eliangiringa Kaale.

**Data curation:** Margareth Ndomondo-Sigonda, Brian Ng'andu.

**Formal analysis:** Margareth Ndomondo-Sigonda, Nelson E. Masota, Eliangiringa Kaale.

**Investigation:** Margareth Ndomondo-Sigonda, Brian Ng'andu.

**Methodology:** Margareth Ndomondo-Sigonda.

**Project administration:** Brian Ng'andu.

**Supervision:** Jacqueline Miot, Shan Naidoo.

**Validation:** Jacqueline Miot, Nancy Ngum.

**Writing – original draft:** Margareth Ndomondo-Sigonda, Nancy Ngum.

**Writing – review & editing:** Jacqueline Miot, Shan Naidoo, Eliangiringa Kaale.

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
