## [Decision Letter · Decision Letter 0]

4 Feb 2020

PONE-D-19-30086

Factors Affecting National Medicines Regulatory Authorities Financial Sustainability –The East African Community Case Study

PLOS ONE

Dear Professor Kaale,

Thank you for submitting your manuscript to PLOS ONE. After careful consideration, we feel that it has merit but does not fully meet PLOS ONE’s publication criteria as it currently stands. Therefore, we invite you to submit a revised version of the manuscript that addresses the points raised during the review process.

We would appreciate receiving your revised manuscript by Mar 20 2020 11:59PM. To enhance the reproducibility of your results, we recommend that if applicable you deposit your laboratory protocols in protocols.io, where a protocol can be assigned its own identifier (DOI) such that it can be cited independently in the future. For instructions see: http://journals.plos.org/plosone/s/submission-guidelines#loc-laboratory-protocols

We look forward to receiving your revised manuscript.

Kind regards,

Sukumar Vellakkal

Academic Editor

PLOS ONE

"Authors would like to express their sincere gratitude to the AUDA-NEPAD for funding this study."

Reviewers' comments:

Reviewer's Responses to Questions

**Comments to the Author**

1. Is the manuscript technically sound, and do the data support the conclusions?

Reviewer #1: Yes

Reviewer #2: Yes

2. Has the statistical analysis been performed appropriately and rigorously? 

Reviewer #1: N/A

Reviewer #2: I Don't Know

3. Have the authors made all data underlying the findings in their manuscript fully available?

Reviewer #1: No

Reviewer #2: Yes

4. Is the manuscript presented in an intelligible fashion and written in standard English?

Reviewer #1: Yes

Reviewer #2: Yes

5. Review Comments to the Author

Reviewer #1: The article covers an important component of the regulatory systems and the conceptual approach to the topic is sound and holds practical relevance to not only African States but to many LMICs. However, few points are identified here which need attention to improve the understanding of the study findings and increase clarity.

Overall Gaps:

Inconsistent use of terms like TFDA and TMDA being used interchangeably throughout the document.

Methodology: The method being mixed method approach has a quantitative and qualitative component of data. However, the number of respondents for each country and their general demographic detail (age, gender, qualification, experience of working, level of engagement with NMRA etc.), an interview or meeting guide need to be shared to support study methodology. The source of primary data collection (e.g, official notifications, reports, webpage of the NMRA etc) is missing (see Page 5, line 105-106). It seems that the review of NMP was a component of methodology as discussed latter in results (subheading 1) but needs to be mentioned here with the sources accessed for retrieving data on funds and policies. The exchange rate applied needs to be mentioned properly “prevailing average annual USD currency exchange rate” is insufficient to serve the purpose.

Results:

Subheading 1

Page 6, line 127-128: Is the financial autonomy laid down in the NMP (please mention clearly)

Subheading 2

Page 7, line 129-131: Needs clear representation of data to support this. It will be nice to adda table with description of each country’s information under financiall autonomy status (Y/N), if Y since (year), under law or rule (State the rule/act or law reference), year of passing and enactment of law.

Subheading 3

Page 7, line 142-144: The figure represents the annual trend only so it can be described in a separately.

Subheading 4

Page 8, line 166: PPA (Please expand abbreviation on first use)

Page 8, line 167-170: Please rephrase to more concise .... 73.20% and 98.25%, respectively over the reported duration

Page 9, line 178-182: It will be nicer to mention the funding agencies contributing up to a certain amount limit or above it for each country to make it more quantifiable to relate to the figure 4.

Subheading 5

Page 9, line 192: PPB ??

Page, line 194-195. Addition of a figure on annual trend need will make the results more clear nd understandable.

Page 9, line 196-200: The point can be shifted to limitations of the study. Moreover, what could have been easily added is the population size and the year wise and/or country wise HR strength of the technical and nontechnical personnel in the NMRA as something that can explain the expenditure.

Discussion

Reference 6 is closely related to the objective and design of the study and is recommended to be discussed the results of the study.

Page 10, line 218: Reference 7 is unpublished, Authors should try to used published references for the details discussed here (references for improved regulatory systems in the countries under study for the study period)

Conclusion:

Lapses in timely release of funds is not mentioned in the results nor data extraction from the interviews or meetings was disclosed in the article which could support this conclusion. Results need to be more explicit to support qualitative segment.

References:

Reference formatting needs attention for uniform style.

Reviewer #2: 1. Insufficient detail on methods – how was data analyzed?

E.g. NMPs- was there a document analysis?

2. Are the NMP stipulations legally binding? The discussion indicates this, but it is not clear.

3. Line 146- 148 says each budget approx. doubled, but it is difficult to discern this about the ZFDA in Fig. 1. Suggest editing figure to include smaller intervals on Y axis.

4. Throughout the paper, “N” is used to denote a USD amount. Please double check whether the use is correct, as I am not familiar with the use of N for a dollar amount (rather only for total population or sample).

5. It should be noted why there is no data for Kenya and Rwanda for Fig. 2.

6. Can the scale for Fig. 3 also be changed to capture the low amounts better?

7. While donor contributions to NMRAs may vary, it would be interesting to know which of the donors provided the most funding, per country and perhaps overall.

8. Fig. 4- It looks as though the SDs are negative. I am not a statistician, but I do not think that’s possible? Suggest a basic statistical review and/or make this more clear in the figure.

9. Line 217- Please spell out QMS and IMS.

10. Please review the grammar in the discussion.

11. Line 242- use of the word “dispose” seems awkward, consider alternative such as “use the revenue”.

12. In lieu of pharmaceutical market size data, one could possibly show, as some kind of possible indication of market size perhaps, the number of medicines listed on the national essential medicines lists. This data is available here: https://global.essentialmeds.org/dashboard/countries

If used, this publication can be cited for the source:

Persaud, N., Jiang, M., Shaikh, R., Bali, A., Oronsaye, E., Woods, H., & al., e. (2019). Similarities and Differences in Essential Medicines Lists of 137 Countries: a quantitative analysis. . Bull World Health Organ.

13. In the discussion, the need for government investment is noted, but I think the opportunities to collect revenue from industry should also be stated as a key source again. Given the projections in market expansion, it seems only fair that industry should not only profit but also be contributing to NMRA activities from which they will ultimately benefit.

14. State opportunities with the African Medicines Regulatory Authority?

15. It seems that paragraph lines 277-284 should be in the discussion rather than conclusion (see comment above on industry funding in discussion). This is too much detail for the conclusion.

16. References require proper formatting.

17. Are there more references to cite on effectiveness of regulatory harmonization?

6. PLOS authors have the option to publish the peer review history of their article (what does this mean?). If published, this will include your full peer review and any attached files.

Reviewer #1: No

Reviewer #2: No

---

## [Author Response · Author response to Decision Letter 0]

4 Apr 2020

Reviewer 1 Comment

Inconsistent use of terms like TFDA and TMDA being used interchangeably throughout the document.

Author’s response 

TFDA has been replaced with TMDA throughout the document

Reviewer 1 Comment

Methodology: 

The method being mixed method approach has a quantitative and qualitative component of data. However, the number of respondents for each country and their general demographic detail (age, gender, qualification, experience of working, level of engagement with NMRA etc.), an interview or meeting guide need to be shared to support study methodology. 

Author’s response 

Added the following in methodology: “Data from six NMRAs was collected through a combination of semi-structured interviews, questionnaires and checklists for the period 2011/12-2014/15 while 2010/11 data served as baseline. Interviews were conducted with heads of NMRAs and monitoring and evaluation experts of the respective agencies” Page 2, lines 28-31; and Pages 5-6, lines 105-

Reviewer 1 Comment

The source of primary data collection (e.g, official notifications, reports, webpage of the NMRA etc) is missing (see Page 5, line 105-106). 

Author’s response 

As stated above, the sources of primary data collection have been added on the methodology section:

Data from six NMRAs was collected through a combination of semi-structured interviews, questionnaires and checklists for the period 2011/12-2014/15 while 2010/11 data served as baseline. 

Interviews were conducted with heads of NMRAs and monitoring and evaluation experts of the respective agencies Pages 5-6, lines 105-109

Reviewer 1 Comment

It seems that the review of NMP was a component of methodology as discussed latter in results (subheading 1) but needs to be mentioned here with the sources accessed for retrieving data on funds and policies. 

Author’s response 

Added in methodology the following text: 

“as shown under indicators guidance notes S2 Table”,

to reflect the indicators used for analysis such as ‘NMRA funding policy’

Changed the title of subheading in results to read 

“Funding Policies and Laws of NMRAs”

Page 6, lines 119-120

Page 7, line 132

Reviewer 1 Comment

The exchange rate applied needs to be mentioned properly “prevailing average annual USD currency exchange rate” is insufficient to serve the purpose.

Author’s response 

The phrase has been omitted from the abstract. 

The ranges of average annual exchange rates over the studied period are now included. 

The authors think it will be more confusing to include the annual average for each country over six years, lowest and highest rates during this period are given instead. Page 6, lines 113 - 116

Reviewer 1 Comment

Results:

Subheading 1:

Page 6, line 127-128: Is the financial autonomy laid down in the NMP (please mention clearly)

Author’s response 

While national medicines policies provide an overarching government commitment to establish NMRAs, it is the medicines law which provide financial autonomy. 

Medicines laws provide for legal power for NMRAs to collect and utilize fees. I have added medicines laws in the sub-heading and text to reflect this

Added a phrase “national medicines policies (NMPs) and medicines laws” for clarity

Page 7, line 133-135

Reviewer 1 Comment

Subheading 2

Page 7, line 129-131: Needs clear representation of data to support this. It will be nice to add a table with description of each country’s information under financial autonomy status (Y/N), if Y since (year), under law or rule (State the rule/act or law reference), year of passing and enactment of law

Author’s response 

Included Table 1 on the level of NMRA autonomy in the EAC Partner States with respective laws and year of enactment and/or amendment Page 7 line 156-158

Reviewer 1 Comment

Subheading 3

Page 7, line 142-144: The figure represents the annual trend only so it can be described in a separately.

Author’s response 

Figure 1 and 2 are now described separately (under two subheadings) Page 7-8, lines 159 - 179

Reviewer 1 Comment

Subheading 4

Page 8, line 166: PPA (Please expand abbreviation on first use)

Page 8, line 167-170: Please rephrase to more concise .... 

73.20% and 98.25%, respectively over the reported duration.

Page 9, line 178-182: It will be nicer to mention the funding agencies contributing up to a certain amount limit or above it for each country to make it more quantifiable to relate to the figure 4.

Author’s response 

The abbreviation PPB (Pharmacy and poisons Board) is used and explained for the first time in Page 7 line 135

Rephrased for clarity: “On average, industry fees contributed up to 73.20% and 98.25% of the total annual revenue for TMDA NDA respectively.”

Individual contribution of each funding partner was not within the scope of the collected data. The data on all funds from these agencies were obtained collectively.

Page 7, line 137

Page 9, line 186-188

Reviewer 1 Comment

Subheading 5

Page 9, line 192: PPB??

Page, line 194-195. Addition of a figure on annual trend need will make the results clearer and more understandable.

Page 9, line 196-200: The point can be shifted to limitations of the study. Moreover, what could have been easily added is the population size and the year wise and/or country wise HR strength of the technical and nontechnical personnel in the NMRA as something that can explain the expenditure.

Author’s response 

The abbreviation PPB (Pharmacy and poisons Board) is used and explained for the first time in Page 7 line 132

Shifted the following text to limitations of the study: 

“In this study, data on population size, the number of pharmaceutical establishments and the pharmaceutical market size was intended to determine the level of investment (in terms of financial and human resources) required to control the market. However, only Burundi provided data on pharmaceutical market size making analysis and comparison between countries difficult”.

A figure capturing the trends of human resource for all NMRAs has been selected (Figure 5).

Page 7, line 137

Pages 14, lines 301-311

Reviewer 1 Comment

Discussion

Reference 6 is closely related to the objective and design of the study and is recommended to be discussed the results of the study.

Author’s response 

Reference 6 is not retrievable from a peer reviewed journal or any other internet accessible source (unpublished). It is therefore dropped, and other references are used in discussion of the study

Reviewer 1 Comment

Page 10, line 218: Reference 7 is unpublished, Authors should try to used published references for the details discussed here (references for improved regulatory systems in the countries under study for the study period)

Author’s response 

Changed reference and replaced with ‘The Compendium of Quality Management System (QMS) Technical Documents for Harmonization of Medicines Regulation in the East African Community’ which was published in 2014 (Ref number 17) Page 11, line 231-234

Reviewer 1 Comment

Conclusion:

Lapses in timely release of funds is not mentioned in the results nor data extraction from the interviews or meetings was disclosed in the article which could support this conclusion. Results need to be more explicit to support qualitative segment

Author’s response 

Deleted ‘timely release of funds’ Page 15, line 320

Reviewer 1 Comment

References:

Reference formatting needs attention for uniform style.

Author’s response 

All references have been formatted as per the PLOS ONE style Pages 16 – 19, Lines 346-407

Reviewer 2 Comment

Insufficient detail on methods – how was data analyzed?

E.g. NMPs- was there a document analysis?

Author’s response 

Added the following in methodology: 

“Quantitative data were analyzed for means and standard deviation using Microsoft Excel©, whereas document analysis was used to extract organize and interpret data from policies and laws”

Page 6, lines 121-124.

Reviewer 2 Comment

Are the NMP stipulations legally binding? The discussion indicates this, but it is not clear.

Author’s response 

While national medicines policies provide an overarching government commitment to establish NMRAs, it is the medicines law which provide financial autonomy. 

Medicines laws provide for legal power for NMRAs to collect and utilize fees. 

We have now added medicines laws in the sub-heading and text to reflect this Page 7, line 132

Reviewer 2 Comment

Line 146- 148 says each budget approx. doubled, but it is difficult to discern this about the ZFDA in Fig. 1. Suggest editing figure to include smaller intervals on Y axis.

Author’s response 

Figures 1 – 4 have been edited to expand the y-axis. However, due to big differences between highest and lowest values, it was difficult to do further expansion than in the current state. Figures 1 – 4.

Reviewer 2 Comment

Throughout the paper, “N” is used to denote a USD amount. Please double check whether the use is correct, as I am not familiar with the use of N for a dollar amount (rather only for total population or sample).

Author’s response 

Deleted ‘N’ associated with $ sign and replaced the dollar sign with USD throughout the text Throughout the text

Reviewer 2 Comment

It should be noted why there is no data for Kenya and Rwanda for Fig. 2.

Author’s response 

The phrase “However, data from Kenya and Rwanda under this aspect was not available” has been included. Page 9, line 175

Reviewer 2 Comment

Can the scale for Fig. 3 also be changed to capture the low amounts better?

Author’s response 

The scale of figure 3 on the y-axis has been changed to better capture the low amounts. Figure 3

Reviewer 2 Comment

While donor contributions to NMRAs may vary, it would be interesting to know which of the donors provided the most funding, per country and perhaps overall.

Author’s response 

Difficult to quantify in a meaningful way. Individual contribution of each funding partner was not within the scope of the collected data. The data on all funds from these agencies were obtained collectively

Reviewer 2 Comment

Fig. 4- It looks as though the SDs are negative. I am not a statistician, but I do not think that’s possible. Suggest a basic statistical review and/or make this clearer in the figure.

Author’s response 

This was possible due to the high variability in annual expenditures of some aspects. Some standard deviation values were therefore larger than the means, which is a possible occurrence. However, to avoid confusions, we have conducted a statistical review and omitted the outliers. See revised figures 3 and 4.

Reviewer 2 Comment

Line 217- Please spell out QMS and IMS.

Author’s response 

Included… ‘quality management systems for QMS’ and dropped IMS

Text now reads: “The assumption is that the increase in revenue could be associated with the improvement in registration processes especially following harmonization of registration systems and the introduction of quality management systems among the EAC Partner States” Page 11, line 233 

Reviewer 2 Comment

Author’s response 

Reviewer 2 Comment

Please review the grammar in the discussion.

Author’s response 

The grammar in the discussion has been reviewed. Page 11-14

Reviewer 2 Comment

Line 242- use of the word “dispose” seems awkward, consider alternative such as “use the revenue”.

Author’s response 

Agreed, the phrase has been changed to “to use the revenue they collect” Page 12, lines 265-266

Reviewer 2 Comment

In lieu of pharmaceutical market size data, one could possibly show, as some kind of possible indication of market size perhaps, the number of medicines listed on the national essential medicines lists. This data is available here: 

https://global.essentialmeds.org/dashboard/countries

If used, this publication can be cited for the source:

Persaud, N., Jiang, M., Shaikh, R., Bali, A., Oronsaye, E., Woods, H., & al., e. (2019). Similarities and Differences in Essential Medicines Lists of 137 Countries: a quantitative analysis. . Bull World Health Organ.

Author’s response 

The reference and link provide global status of essential medicines list per country which may not necessarily give an indication of the pharmaceutical market size

Reviewer 2 Comment

In the discussion, the need for government investment is noted, but I think the opportunities to collect revenue from industry should also be stated as a key source again. Given the projections in market expansion, it seems only fair that industry should not only profit but also be contributing to NMRA activities from which they will ultimately benefit

Author’s response 

Agreed, substantiated in the discussion Page 12, lines 268-269

Reviewer 2 Comment

State opportunities with the African Medicines Regulatory Authority?

Author’s response 

Added text on the African Medicines Regulatory Harmonization (AMRH) Initiative as a foundation for African Medicines Agency (AMA) in the discussion, also included Africa Continental Free Trade Area (AfCFTA) as another opportunity Pages 13-14, lines 285-299

Reviewer 2 Comment

It seems that paragraph lines 277-284 should be in the discussion rather than conclusion (see comment above on industry funding in discussion). This is too much detail for the conclusion.

Author’s response 

Agreed, the paragraph in lines 277 – 284 has been removed from the conclusion and moved to the discussion section together with some rephrasing Page 12, lines 254-261

Reviewer 2 Comment

References require proper formatting.

Author’s response 

All references have been formatted as per the PLOS ONE style Pages 16 – 18, Lines 346-407

Reviewer 2 Comment

Are there more references to cite on effectiveness of regulatory harmonization?

Author’s response 

Included References 22 and 23 Page 18, line 401 - 407

---

## [Decision Letter · Decision Letter 1]

14 May 2020

PONE-D-19-30086R1

National Medicines Regulatory Authorities Financial Sustainability –The East African Community Case Study

PLOS ONE

Dear Dr Kaale,

Thank you for re-submitting your manuscript to PLOS ONE. This has now been reviewed and we have received advice from both reviewers. Both reviewers have indicated that you have made good progress with the amended version of your manuscript. One of the reviewers have raised a number of issues - we feel if you can address these  the paper should be able to be published by PLOS ONE. Therefore, we invite you to submit a revised version of the manuscript that addresses the points raised during the latest review process.

We would appreciate receiving your revised manuscript by Jun 28 2020 11:59PM. To enhance the reproducibility of your results, we recommend that if applicable you deposit your laboratory protocols in protocols.io, where a protocol can be assigned its own identifier (DOI) such that it can be cited independently in the future. For instructions see: http://journals.plos.org/plosone/s/submission-guidelines#loc-laboratory-protocols

We look forward to receiving your revised manuscript.

Kind regards,

Eduard J Beck, PhD, FAFPHM, FFPH, FRCP

Academic Editor

PLOS ONE

Reviewers' comments:

Reviewer's Responses to Questions

**Comments to the Author**

1. If the authors have adequately addressed your comments raised in a previous round of review and you feel that this manuscript is now acceptable for publication, you may indicate that here to bypass the “Comments to the Author” section, enter your conflict of interest statement in the “Confidential to Editor” section, and submit your "Accept" recommendation.

Reviewer #1: All comments have been addressed

Reviewer #2: (No Response)

2. Is the manuscript technically sound, and do the data support the conclusions?

Reviewer #1: Yes

Reviewer #2: Yes

3. Has the statistical analysis been performed appropriately and rigorously? 

Reviewer #1: Yes

Reviewer #2: I Don't Know

4. Have the authors made all data underlying the findings in their manuscript fully available?

Reviewer #1: Yes

Reviewer #2: No

5. Is the manuscript presented in an intelligible fashion and written in standard English?

Reviewer #1: Yes

Reviewer #2: Yes

6. Review Comments to the Author

Reviewer #1: (No Response)

Reviewer #2: Overall the quality has improved, although it still requires more detail to properly understand steps taken to conduct this study and therefore to evaluate the validity of the findings. I think the discussion is very interesting now!

1. While the title indicates that a case study methodology was used, the methods section does not. In my view the study did not take a case study approach (which is a methodology to intensively study a case for the purpose of understanding a larger population). I suggest to remove case study from the title as it may be misleading and replace it with something like “National Medicines Regulatory Authorities Financial Sustainability in The East African Community”.

2. Abstract Methodology: incorrect use of semi-colon, replace with commas. This is also the case on lines 62-64, 116-119 and a few more throughout the manuscript.

3. Line 99: Reconsider case study use here, consider alternative wording (as per my comment 1).

4. Whether semi-structured interviews were done and how is unclear, as there is no detailed information on participants or data collection in this regard. S1 indicates that respondents completed the questionnaire form on their own or perhaps it was administered by the study team? It appears that a descriptive analysis was done from a questionnaire, but it’s hard to tell. If interviews were done, this requires a lot more detail, such as:

- Was a topic guide used for the semi-structured interviews or was it based directly on the questionnaire (and thus perhaps not really semi-structured)?

- How were participants selected, how many participants responded, what was the length and format of the interviews?

- Were these audio-recorded and transcribed? How was data analyzed (more information so I could reproduce this)?

- How was sampling conducted?

5. In any case, it would help to have more participant information. Were participants all the heads of NMRAs participants and M&E experts (i.e. did all agree in each of the countries?), how many respondents were there total? Why were these specifically selected?

6. Lines 108-109: In the follow-up visits, who provided the missing data at the agencies and who validated the data? How exactly was the data validated (e.g. official records)? Or how do you know it is reliable? Were these different respondents than the initial interviews?

7. Related, where was the budget data obtained from specifically (official records, or did you rely on responses from individuals) and how was it validated if it was through responses?

8. How was the questionnaire/ data collection tool developed? Detail is needed to understand what it is based on (any previously used standardized tools?) and its validity.

9. Line 230-232: Why is this the assumption? Can you be more explicit?

10. Line 234 (and throughout the manuscript): Check capitalizations, as these are inconsistent in some parts of the paper. I also think there is a typo here with “multi-county” instead of “multi-country”.

11. Line 240: pluralize sound government policies and legal frameworks?

12. Line 247: typo with t before government?

13. Line 271: “chronic lifestyle diseases” implies the conditions are a result solely of individual behaviours which is shown to be generally not true, suggest changing to non-communicable diseases or simply chronic conditions

14. Line 314: “in particular” isn’t necessary here

15. PREVIOUS COMMENT

Reviewer 2 Comment

Throughout the paper, “N” is used to denote a USD amount. Please double check whether the use is correct, as I am not familiar with the use of N for a dollar amount (rather only for total population or sample).

Author’s response

Deleted ‘N’ associated with $ sign and replaced the dollar sign with USD throughout the text Throughout the text

REVIEWER REPLY: One ‘N’ was not deleted/missed on page 9.

7. PLOS authors have the option to publish the peer review history of their article (what does this mean?). If published, this will include your full peer review and any attached files.

Reviewer #1: No

Reviewer #2: No

---

## [Author Response · Author response to Decision Letter 1]

23 May 2020

Reviewer 2 Comment

Have the authors made all data underlying the findings in their manuscript fully available? 

Author’s response 

Agreed, the data underlying the findings described in the manuscript have been availed as Supporting information. (S2)

Reviewer 2 Comment

While the title indicates that a case study methodology was used, the methods section does not. In my view the study did not take a case study approach (which is a methodology to intensively study a case for the purpose of understanding a larger population). I suggest to remove case study from the title as it may be misleading and replace it with something like “National Medicines Regulatory Authorities Financial Sustainability in The East African Community”. Authorities Financial Sustainability in The East African Community”. (Line 1-2)

Author’s response 

We agree with the proposal. The title has been changed to read “National Medicines Regulatory

Reviewer 2 Comment

Abstract Methodology: incorrect use of semi-colon, replace with commas. This is also the case on lines 62-64, 116-119 and a few more throughout the manuscript.

Author’s response 

Agreed, incorrectly used semi-colons have been replaced with comas (Lines 38-44, 61-64, 137-139 and throughout the manuscript)

Reviewer 2 Comment

Line 99: Reconsider case study use here, consider alternative wording (as per my comment 1).

Author’s response 

The title has been changed to read “National Medicines Regulatory Authorities Financial Sustainability in the East African Community” (Line 1-2)

Reviewer 2 Comment

Whether semi-structured interviews were done and how is unclear, as there is no detailed information on participants or data collection in this regard. S1 indicates that respondents completed the questionnaire form on their own or perhaps it was administered by the study team? It appears that a descriptive analysis was done from a questionnaire, but it’s hard to tell. If interviews were done, this requires a lot more detail, such as:

- Was a topic guide used for the semi-structured interviews or was it based directly on the questionnaire (and thus perhaps not really semi-structured)?

- How were participants selected, how many participants responded, what was the length and format of the interviews?

- Were these audio-recorded and transcribed? How was data analyzed (more information so I could reproduce this)?

- How was sampling conducted? 

Author’s response 

Agreed, the following paragraphs have been added on the methodology section to provide more details about tools, mode and participants involved data collection:

“The first phase of data collection involved self-administration of the questionnaire and checklists (S1) by the selected informants. These included the NMRA’s head, one monitoring and evaluation personnel from each NMRA and a project officer from the EAC MRH project, making a total of 13 respondents. All informants were purposefully selected based on their roles and participation in medicines policy and regulation of medical products in the respective NMRA.” (lines 112 -117)

“Semi-structured interviews, also involving the above-mentioned respondents, were conducted following the successful completion of the questionnaire and checklists data collection phase. In addition, one NMRA staff responsible for medicines registration, GMP inspections, legal affairs, human resource and finance were also interviewed in each NMRA. An invitation letter and interview topic guide were sent to the interviewees in advance. Interviews were conducted on face to face basis, each session lasting for 1 to 2 hours. Responses were recorded by means of selective written notes, which were thereafter subjected to qualitative analysis. Follow up visits were conducted aiming at collecting the missing data and validating the previously collected data” (Lines 118 – 126)

Reviewer 2 Comment

In any case, it would help to have more participant information. Were participants all the heads of NMRAs participants and M&E experts (i.e. did all agree in each of the countries?), how many respondents were there total? Why were these specifically selected? 

Author’s response 

Yes, all heads of NMRAs and M&E experts from each country agreed to participate in the study. We have included the total number (13) of participants in questionnaire and checklist in the above response. Moreover, during semi-structured interviews 5 additional participants were interviewed. These were the staff responsible for medicines registration, GMP inspections, legal affairs, human resource and finance (also stated in the above response). 

We selected all respondents based on their prevailing roles within the NMRA. This meant they were well informed on all aspects of relevance to the objectives of this study, hence enabling availability of more detailed and reliable information

Reviewer 2 Comment

Related, where was the budget data obtained from specifically (official records, or did you rely on responses from individuals) and how was it validated if it was through responses? 

Author’s response 

Budget data was obtained from official records, heads of NMRAs signed consent form to make data available and agreed on publication of the outcome of research.

Reviewer 2 Comment

How was the questionnaire/ data collection tool developed? Detail is needed to understand what it is based on (any previously used standardized tools?) and its validity.

Author’s response 

“Questionnaire and checklists were developed based on the information obtained from the preliminary situational analysis study. Moreover, additional questions were adopted from the WHO Global benchmarking tool for evaluation of national regulatory systems. Validation of the questionnaire and checklists was done in pilot study involving 2 NMRAs.” (Lines 108 – 111)

Reviewer 2 Comment

Line 230-232: Why is this the assumption? Can you be more explicit? 

Author’s response 

This assumption is based on findings from another unpublished study we conducted aiming at assessing the medicines registration and quality management systems in the East African Community, where we observed that the NMRAs which showed increased efficiency in marketing authorization process after the harmonization of registration systems.

For better clarity, we have rephrased the statement to read:

“The increase in revenue could be associated with the observed improvement in registration processes following the harmonization of registration systems and the introduction of quality management systems among the EAC partner States”. (Lines 247 -249)

Reviewer 2 Comment

Line 234 (and throughout the manuscript): Check capitalizations, as these are inconsistent in some parts of the paper. I also think there is a typo here with “multi-county” instead of “multi-country”.

Author’s response 

Agreed, capitalizations checked and corrected throughout the manuscript. The typo in “multi-county” has been corrected to “multi-country”. (line 250 and throughout the manuscript)

Reviewer 2 Comment

Line 240: pluralize sound government policies and legal frameworks?

Author’s response 

Agreed, the policy and framework has been pluralized to read “policies and legal frameworks”. (Line 256)

Reviewer 2 Comment

Line 247: typo with t before government?

Author’s response 

Agree, the t has been omitted (line 263)

Reviewer 2 Comment

Line 271: “chronic lifestyle diseases” implies the conditions are a result solely of individual behaviors which is shown to be generally not true, suggest changing to non-communicable diseases or simply chronic conditions

Author’s response 

Agreed, changed to read “chronic diseases”. (Line 287)

Reviewer 2 Comment

Line 314: “in particular” isn’t necessary here 

Author’s response 

Agreed, “in particular has been deleted. (Line 329)

Reviewer 2 Comment

Throughout the paper, “N” is used to denote a USD amount. Please double check whether the use is correct, as I am not familiar with the use of N for a dollar amount (rather only for total population or sample).

Author’s response

Deleted ‘N’ associated with $ sign and replaced the dollar sign with USD throughout the text Throughout the text

REVIEWER REPLY: One ‘N’ was not deleted/missed on page 9. 

Author’s response 

Agreed, the remaining ‘N’ has been deleted (Line 188)

---

## [Editor Report · Decision Letter 2]

7 Jul 2020

National Medicines Regulatory Authorities Financial Sustainability –The East African Community Case Study

PONE-D-19-30086R2

Dear Dr Eliangiringa A. Kaale

We’re pleased to inform you that your manuscript has been judged scientifically suitable for publication and will be formally accepted for publication once it meets all outstanding technical requirements.

Kind regards,

Helen Schneider, MBChB, MMed, PhD

Academic Editor

PLOS ONE
---

## [Editor Report · Acceptance letter]

9 Jul 2020

PONE-D-19-30086R2 

National Medicines Regulatory Authorities Financial Sustainability in the East African Community 

Dear Dr. Kaale:

I'm pleased to inform you that your manuscript has been deemed suitable for publication in PLOS ONE. Congratulations! Your manuscript is now with our production department. 

Kind regards, 

on behalf of

Dr. Helen Schneider 

Academic Editor

PLOS ONE